# A Sweet Warning: Mucin-Type O-Glycans in Cancer

**DOI:** 10.3390/cells11223666

**Published:** 2022-11-18

**Authors:** Yuhan Zhang, Lingbo Sun, Changda Lei, Wenyan Li, Jiaqi Han, Jing Zhang, Yuecheng Zhang

**Affiliations:** 1Medical College of Yan’an University, Yan’an University, Yan’an 716000, China; 2Department of Gastroenterology, Ninth Hospital of Xi‘an, Xi’an 710054, China; 3Key Laboratory of Analytical Technology and Detection of Yan’an, College of Chemistry and Chemical Engineering, Yan’an University, Yan’an 716000, China

**Keywords:** mucin-type o-glycosylation, cancer, tumor-associated antigens antigen, tumor therapy

## Abstract

Glycosylation is a common post-translational modification process of proteins. Mucin-type O-glycosylation is an O-glycosylation that starts from protein serine/threonine residues. Normally, it is involved in the normal development and differentiation of cells and tissues, abnormal glycosylation can lead to a variety of diseases, especially cancer. This paper reviews the normal biosynthesis of mucin-type O-glycans and their role in the maintenance of body health, followed by the mechanisms of abnormal mucin-type O-glycosylation in the development of diseases, especially tumors, including the effects of Tn, STn, T antigen, and different glycosyltransferases, with special emphasis on their role in the development of gastric cancer. Finally, tumor immunotherapy targeting mucin-type O-glycans was discussed.

## 1. Introduction

Glycans are the most information dense biological macromolecules in animal cells, and glycosylation is the most common protein post-translational modification (PTM) process in eukaryotes, in which more than 60% of proteins are linked to carbohydrates [1]. Glycans, like nucleic acids, proteins, lipids, and metabolites, are ubiquitous in nature and are the most diverse and rapidly developing class of significant molecules. Compared with other cellular macromolecules such as DNA/RNA and protein, the expression mode of glycans is completely different. Nucleotides and proteins are linear polymers with only one basic type of bond, whereas a monosaccharide can produce either an alpha or beta bond to any of several positions on another monosaccharide. Thus, three different nucleotides or amino acids can only produce six trimers, but theoretically, three different monosaccharides can produce a large number of different polysaccharides. Moreover, the richer structural diversity of glycans leads to the technical difficulty of analyzing these molecules, which makes the research on glycans so far lagged on nucleic acids and proteins [2,3]. Protein glycosylation is a series of enzymatic reactions that occur in the ER and the Golgi apparatus, and more than 200 glycosyltransferases are involved to determine the initiate locations of glycans on proteins, as well as the extension and branching structures of glycans [4,5]. Different types of glycoconjugates, including N and O linked glycoproteins, glycolipids, proteoglycans (PGs), and glycosylphosphatidylinositol (GPI) anchored proteins are discovered in human glycome [5]. Two main types (>95%) of glycans present in serum are N-glycans and O-glycans [6]. In O-glycosylation, sugars are added to the oxygen atoms of the hydroxyl groups of serine or threonine residues in mammals which are composed of ten monosaccharides: fucose (Fuc), galactose (Gal), glucose (Glc), N-acetylglucosamine (GlcNAc), N-acetylgalactosamine (GalNAc), glucuronic acid (GlcA), iduronic acid (IdoA), sialic acid (Sia), mannose (Man), and xylose (Xyl) [5]. There are several types of O-glycosylation, but the most common and diverse one is the O-GalNAc glycosylation, also called mucin-type O-glycosylation, which starts in the Golgi apparatus and is abundant in mucins [7]. Mucins contain a class of high molecular weight membrane-bound and secreted glycoproteins; the secreted mucin family includes MUC2, MUC5Ac, MUC5B, MUC6, MUC7, MUC8, MUC12, and MUC19. The membrane-bound mucin family includes MUC1, MUC3A, MUC3B, MUC4, MUCH-13, MUC15-17, MUC20, MUC21, and MUC22, which possess a large number of O-glycosylation sites, usually composed of tandem repeats, and more than 80% of cell membrane proteins and extracellular secreted proteins are O-GalNAc sugars sialylated protein [8]. Moreover, mucin-like regions have also been identified on the envelope glycoproteins of many bacteria and viruses, characterized by a high density of localized O-glycosylation. For example, herpes simplex virus type 1 (HSV-1) exhibits mucin-like regions on its glycoprotein C, resulting in a structure similar to cell surface glycans [9].

Although mucin-type N-glycosylation has been less studied than O-glycosylation, N-glycosylation plays an important role in the biosynthesis, folding, stability, function, and cell surface localization of mucin. By interacting with lectin-like chaperones, N-glycans ensure that proteins keep proper folding before leaving the ER [10]. In addition, there is an inextricable relationship between N- and O-glycosylation, and some shared carbohydrate antigens are formed through the linkage between sugar chains. For example, sialyl Lewis a (NeuAcα2-3Galβ1–3[Fucα1-4] GlcNAc, SLe^a^) and sialyl-Lweis x (NeuAcα2-3Galβ1–4[Fucα1-3]GlcNAc, SLe^x^) are typical tumor-associated antigens occurring on both N- and O- glycans, which are formed by sialic acid and fucose under either type 1 (Galβ1-3GlcNAc) or type 2 (Galβ1-4GlcNAc) chains. [11]. The N-acetylgalactosaminyltransferases β4GalNAcT3 and β4GalNAcT4 can transfer GalNAc to GlcNAc residue in both N- and O-glycans to generate LacdiNAc antigen, which is also a valid biomarker for different cancers or even different stages of certain cancers [12].

Glycosylation is involved in various in vivo reaction processes, including but not limited to immune responses, protein synthesis and degradation, signal transduction, cell differentiation, and cell-cell interactions [4]. Mucin-type O-glycosylation can regulate protein stability and is essential for the normal development, growth, and differentiation of cells and tissues [13,14]. One of its important functions is to mediate mutual recognition between proteins [15]. Mucin-type O-glycans can serve as substrates for nonenzymatic sugar-binding proteins. For example, selectins and hemagglutinins are two types of lectins located in the leukocyte-vasculature system, which bind to sugar epitopes and induce signal transduction between cells and thereby affecting many key response processes, including cell growth, apoptosis, endocytosis, cell-cell interactions, cell-matrix interactions, matrix network assembly, oocyte fertilization, etc. [15,16]. Mucin-type O-glycosylation is also important for water binding, which is often found on outer surfaces that lack a hydrophilic layer, such as those of the digestive tract, genitals, and respiratory tract; these mucins carry sialylated sugar chains that are able to create regions with a large negative charge that allow mucins to bind large amounts of water and form mucus [15]. The main function of these mucin solutions and gels is to form a protective layer with antibacterial properties against bacteria, strong acids, alkalis, and some mechanical damage [17]. For example, the mucin MUC2 provides protection for the intestinal epithelium and can inhibit the occurrence of colitis and colorectal cancer [18].

Abnormal mucin-type O-glycosylation is associated with many human diseases, including COVID-19 and tumor development [19]. Abnormalities in the mucin-type O-glycan synthetic pathway under pathological conditions can lead to major cellular glycosylation changes and key changes in biological processes, ultimately leading to and regulating the course of various diseases [20]. For example, abnormal mucin-type O-glycosylation in glomeruli leads to abnormal expression of the Tn antigen (GalNAc-Ser/Thr), which induces the production of autoantibodies against IgA1 and subsequently leads to the formation and deposition of a large number of IgA1 immune complexes in the glomeruli, resulting in kidney injury [21]. In addition, altered mucin-type O-glycosylation can affect the aggressiveness of tumor cells, including the ability to spread through the circulation and metastasize distantly [22]. Aberrant glycosylation in cancer is the incomplete biosynthesis of mucin-type O-glycans that accumulate to form immature truncated O-glycans, such as Tn antigen [5]. This review will focus on the mucin-type O-glycans correlation between tumorigenesis and progression.

## 2. Biosynthetic Pathway of Mucin-Type O-Glycans in Humans

The first step in mucin-type O-glycosylation is the linking of UDP-GalNAc to Ser/Thr residues of the protein to form the initial structure (Tn antigen) in the presence of a large family of homologous polypeptide GALNT enzymes. The initial structure is then further processed by the addition of different monosaccharides to form the core structure and the extended structure [7]. There are 20 unique *GALNT* isozyme genes in the human genome, and these isozymes have distinct and partially overlapping specificities. Most of them have a catalytic structural domain and a lectin structural domain that form effective modifications of specific Ser/Thr residues by binding to the product [23,24].

After the Tn antigen is formed in the Golgi, there are at least three ways of downward extension. The first is to synthesize the core 1 structure (Galβ1-3GalNAcαSer/Thr, designed T antigen) through the core 1 β1,3-galactosyltransferase (C1GALT1); the second is through the core 3 β1,3-N-acetylglucosamine transferase (β3Gn-T6) synthesizes the core 3 structure (GlcNAcβ1-3GalNAcαSer/Thr); the third is synthesized STn antigen (Neu-Acα1-6GalNAcαSer/Thr) through sialyltransferase ST6GalNAc-I. The most common modification of Tn antigen is under the action of C1GALT1 to generate T antigen [25]. These core structures are often further extended and capped by sialic acid or fucose residues (Figure 1). Notably, the region of occurrence of Tn antigen generation catalyzed by the homologous polypeptide GALNT enzymes and T antigen generation catalyzed by C1GALT1 may be different, as co-localization of GALNTs and C1GALT1 in the same region may affect the efficiency of mucin-type O-glycosylation [7,26]. Moreover, C1GALT1 is a special enzyme because its active expression depends on the coexpression of its molecular chaperone, COSMC, which will be described in more depth later [27].

## 3. The Role of Mucin-Type O-Glycans in Health and Diseases

The importance of glycosylation in various physiological and pathological processes, and in the development of biopharmaceuticals over the past decade cannot be overstated [29]. Normal biosynthesis of mucin-type O-glycans is associated with body health and internal environmental homeostasis, whereas abnormal mucin-type O-glycans are related to the development of cancer and other pathological processes [30]. More than 100 rare congenital disorders of glycosylation (CDG) are caused by defects in genes that regulate glycosylation, 50 of which are caused by defects in the glycosyltransferase genes [31]. Moreover, the GALNTs are the largest family of glycosyl-transferases, which have been shown to play an important role in human health and diseases. Sun et al., induced a mouse renal fibrosis model in C57BL/6 mice with unilateral ureteral obstruction (UUO) and folic acid (FA) in vivo. Over time, in vivo experiments showed that GALNT1 expression was downregulated, let-7i-5p expression was upregulated, and the elevation of serum IL-6, IL-1β, and TNF-α levels. In vitro experiments showed that the expression level of let-7i-5p was negatively correlated with GALNT1 and that overexpression of GALNT1 could inhibit inflammation induced by let-7i-5p, thereby suppressing the development of renal fibrosis in mice [32]. *GALNT3* mutations lead to inactivated cleavage of FGF23 and dysregulation of phosphate homeostasis resulting in familial tumoral calcinosis (FTC), characterized by hyperphosphatemia, altered bone density, and the development of subcutaneous calcified tumors; whereas under normal conditions, GALNT3 mediated FGF23 glycosylation protects FGF23 from protease cleavage [33,34]. Pelus et al. also found that a loss of GALNT3 inhibited the glycosylation of MUC10, thereby altering the composition and stability of the oral microbiota [35].

After the Tn antigen is generated in the first step, Gal is added at the β1,3 junction through another family of enzymes (C1GALTs) [36]. While only one C1GALT1 exists in mammals, the other nine potential members exist in drosophila [37]. In humans, the C1GALT1 encoded protein transfers Gal from UDP-Gal to GalNAc-Ser/Thr (Tn antigen), resulting in T antigen, which is the basis for the formation of complex O-glycans such as the core 2 structure and ST antigen [38]. Xia et al. found that the deletion of *C1GALT1* in mice resulted in the ubiquitous expression of the Tn antigen [39]. In addition, researchers found that Jurkat cells express Tn antigen and lack C1GALT1 activity. Surprisingly, Jurkat cells contain normal levels of *C1GALT1* transcripts, even though the protein expression levels are very low, which means that the C1GALT1 protein is degraded. Then, they found the degradation of the C1GALT1 protein is not a mutation of its own gene but is called a mutation in the X-linked genes of *COSMC* [27]. Although COSMC lacks galactosyltransferase activity, it is a molecular chaperone necessary for C1GALT1 to fold correctly and exert its activity. Early studies of COSMC function incorrectly suggested that the COSMC gene might be a second C1GALT1 [40]. In the ER, COSMC cotranslates and interacts with inactive C1GALT1 to generate active C1GALT1, which is then transported to the Golgi apparatus to function [41].

C1GALT1 and COSMC play important roles in the occurrence and development of different tumors, with both oncogenic and procancer effects. C1GALT1 is highly expressed in head and neck squamous cell carcinoma (HNSCC), breast cancer (BC), esophageal carcinoma (ESCA), hepatocellular carcinoma (HCC), and laryngocarcinoma, enhancing the malignant phenotype of cancer cells and it is associated with a poor survival rate and poor prognosis of patients [42,43,44]. However, the knockout of *C1GALT1* in colorectal cancer HCT 116 cells significantly promoted cell proliferation, adhesion, migration, invasion, and directly induced epithelial mesenchymal transformation (EMT) in colorectal cancer cells [45]. Huang et al. found that COSMC expression was upregulated in colorectal cancer, which promoted the proliferation, migration, and invasion of colon cancer cells by activating MEK/ERK and PI3K/Akt signaling pathways, promoted the growth of tumor cells and decreased the survival rate of tumor mice, and knocking down *COSMC* in SW480 cells could inhibit these malignant phenotypes [46]. Lee et al. also found that COSMC expression was upregulated in proliferating hemangiomas, which promoted endothelial cell growth and phosphorylation of AKT and ERK in human umbilical vein endothelial cells. Furthermore, COSMC was able to regulate VEGF-triggered phosphorylation of VEGFR2, suggesting that COSMC is a novel regulator of VEGFR2 signaling in endothelial cells [47]. Jiang et al. found that *COSMC* mutation leads to its loss of function, which results in reduced C1GALT1 activity, as well as downregulation of MUC2 expression. Transfection of LS174T cells with WT *COSMC* restored mature O-glycosylation, thereby inhibiting the proliferation, migration, and antiapoptotic capacity of cancer cells. In conclusion, aberrant O-glycosylation, due to *COSMC* mutation, promotes the development of (CRC) by directly inducing the oncogenic properties of cancer cells [48].

## 4. Tumor-Associated Antigens: Tn Antigen, STn Antigen, T Antigen

Abnormal mucin-type O-glycosylation is closely associated with the occurrence and development of tumors [49]. In normal cells, maturation of mucin-type O-glycans is usually elongated and produces branched chains, which are eventually covered by sialic acid or fucose. In contrast, cancer cells express only the intermediate product of early biosynthesis, also known as truncated O-glycans, which are observed in almost all epithelial cell carcinomas. Notably, it is observed very frequently in early epithelial precancerous lesions in which adenocarcinoma develops [50,51].

COSMC acts as a molecular chaperone for C1GALT1 activity and becomes a molecular target for regulating the expression of Tn, STn, and T antigens [52]. Downregulation of *COSMC* expression or mutation could lead to upregulation of Tn antigen and STn antigen expression, resulting in truncated O-glycans, which can directly induce oncogenic characteristics of cancer cells, including enhanced proliferation and invasion, loss of tissue structure, and destruction of basal mucosa [53,54]. The relationship between Tn antigen and tumors were first identified by Springer and colleagues in 1975. When they studied human MN blood group antigens, designed specific monoclonal antibodies against Tn antigen, and found that Tn antigen was upregulated in many tumor cells [55]. Several studies have found a high expression of Tn antigen in 10–90% of tumor samples in bladder, cervical, ovarian, colon, lung, gastric, and prostate tumors [56,57,58]. Furthermore, in a variety of cancers, including lung, cervical, colorectal, gastric, and breast cancers, the overexpression of Tn antigen is associated with metastasis, and potentially is associated with poor prognosis [59]. Subsequently, Springer first suggested in 1984 that the expression levels of T and Tn antigens usually correlate with the degree of tumor differentiation. In addition, at the molecular level, the aggregation of T and Tn antigens on the surface of tumor cells may be associated with an invasive ability [56]. Notably, Tn antigen seems to be expressed mainly in epithelial tumors and less in hematological tumors [51].

Due to their similar structures and synthetic pathways, STn antigen and Tn antigen are often upregulated simultaneously in tumor cells at the same time, and STn antigen expression upregulated has been shown to contribute to tumorigenesis, and is associated with a poor prognosis in cancer patients, including gastric, colon, breast, lung, esophageal, prostate, and endometrial cancers [60]. It is regulated by different mechanisms in different cancers. STn antigen upregulation has been shown to be induced by ST6GalNAc-I in gastric, breast, and prostate cell lines; ST6GalNAc-I is a key enzyme in STn antigen synthesis and has been shown to exert its procancer effects through upregulation of STn antigen in many tumors [61]. For example, Julien et al. found that ST6GalNAc-I regulated induction of STn antigen overexpression enhanced the tumorigenicity of breast cancer cells MDA-MB-231 [62]. Ferreira et al. found that ST6GalNAc-I induced STn expression, promoted cell migration, and invasive ability in the bladder cancer cell line MCR [63]. In addition, STn antigen could protect metastatic cells metastasized via blood channels from degradation in the bloodstream from being degraded by the immune system. A possible mechanism is that STn antigen promotes the spread of tumor cells in vivo by reducing cell and intercellular aggregation. For example, by disrupting the interaction between galactose residues, it promotes the individual cells from the primary tumor site, thereby facilitating the spread of tumor cells in vivo [64,65].

T antigen was discovered in 1920 but not recognized as a tumor antigen until 1975 [59]. STn antigen is a disaccharide, partially covered by carbohydrates and hidden in normal cells, however, it is exposed in cancers such as breast, lung, prostate, and bladder cancers, and its upregulated expression is associated with poor prognosis and poor survival rate in cancer patients [66,67]. Expression of T antigen is not only related to COSMC and C1GALT1, it is also related to the dissipation of pH gradient in the Golgi [68]. Golgi lumen alkalinization frequently occurs in tumor cells, which can reposition certain glycosyltransferases, leading to an abnormal synthesis of mucin-type O-glycans [69]. Recently, Valoskova et al. identified an atypical member of the major facilitator superfamily (MFS), called minervain drosophila melanogaster, which regulates mucin-type O-glycosylation to upregulate T antigen, with possible implications for cancer, providing a new perspective on the study of T antigen [70]. 

In addition, the abnormal mucin-type O-glycosylation observed in cancer is usually accompanied by aberrant expression of mucins because of the large number of O-glycosylation sites on mucins [71]. Mucin plays an extremely important role in the development of cancers. For example, Rowson-Hodel et al. found that MUC4 may enhance the survival of tumor cells in circulation by promoting the binding of circulating tumor cells to blood cells, thereby promoting tumor metastasis [72]. Hsu et al., observed in colon cancer that the downregulation of MUC2 expression caused an increase in IL-6 production, which in turn activated the STAT3 signaling pathway in colon cancer cells and promoted EMT [73]. Hoshi et al. found that MUC5Ac may inhibit neutrophil-mediated antitumor effects by mediating the immune escape of tumor cells, then they investigated the effect of MUC5Ac on TRAIL-mediated apoptosis and found that MUC5Ac inhibited TRAIL-induced apoptosis in human pancreatic cancer cells [74].

## 5. Abnormal Mucin-Type O-Glycosylation in Tumors

The mechanisms regulating aberrant mucin-type O-glycosylation in tumors are not fully understood. The current studies have involved the following points: mutation or dysregulation of glycosyltransferases; repositioning of GALNTs from the Golgi apparatus to the ER, and hypermethylation of *COSMC* (Table 1).

GALNTs act as both a tumor-promoting factor and a tumor-suppressing factor, which may depend on the specific cell or different cancer types. Huang et al., found that the knockdown of *GALNT8* led to the inhibition of the BMP/SMAD/RUNX2 axis, which reduced ERα transcription, thereby inhibiting the proliferation of breast cancer cells [75]. Zheng et al., found that GALNT12 promotes the malignant features of Glioblastoma Multiforme (GBM) by activating the PI3K/Akt/mTOR axis and may serve as a novel prognostic biomarker and potential therapeutic target for GBM [76]. Hu et al., found that GALNT2 activates the PI3K/Akt and MAPK/ERK pathways by altering the O-glycosylation of ITGA5. In vitro experiments showed that the knockdown of the *GALNT2* gene inhibited the proliferation, migration, and invasion of nonsmall lung cancer cells and induced apoptosis and cell cycle arrest. In vivo experiments showed that the knockdown of *GALNT2* inhibited the formation of tumor nodules in nude mice [77]. Contrary to these above, Huang et al. have shown in breast cancer that GALNT8 is a tumor suppressor that inhibits metastasis of breast cancer cells and suppresses EMT by inhibiting the EGFR signaling pathway [78]. Liu et al., found that GALNT2 could suppress the malignant phenotype of gastric adenocarcinoma (GCA) cells by inhibiting MET activity or the EGFR Akt signaling [79,80].

Previously, David et al. found that the expression level of Tn antigen in breast cancer was strongly influenced by GALNTs subcellular localization, and there was no lack of C1GALT1 activity in hundreds of human breast cancer samples with high expression of Tn antigen, but they found that GALNTs relocalization from the Golgi to the ER induced high expression of Tn antigen and promoted cell migration and invasion [82]. Bard et al. in 2017 proposed a possible mechanism by which either depletion of ERK8 or activation of Src could induce significant migration of GALNT1, GALNT2, GALNT3, GALNT4, and GALNT6, and increase the activity of GALNTs in HeLa cells, with activation of Src having a stronger effect on GALNTs. This relocation of GALNTs from the Golgi to the ER leading to increased GALNTs activity is referred to as the GALNTs activation (GALA) pathway [83]. They subsequently found that overexpression of EGFR could enhance ERK8 depletion-induced relocalization, thereby activating the GALA pathway [84]. Furthermore, Chia et al., 2021 proposed that the Golgi to the ER transport depends on the GTP exchange factors GBF1 and Arf-GDP, and they showed that Src facilitates the binding of GBF1 to Arf-GDP by activating two phosphorylation sites on GBF1, Y876, and Y898, thereby promoting the relocalization of GALNTs (Figure 2) [85]. Thomas et al. found that hypermethylation of *COSMC* in pancreatic cancer led to the aberrant expression of Tn and STn antigens. In vitro experiments showed that hypermethylation of *COSMC* promoted the migration and invasive ability of pancreatic cancer cells by inducing EMT and stem cell properties, and in vivo experiments showed that hypermethylation of *COSMC* increased the weight and volume of tumor nodules and decreased the survival rate in mice [81]. As mentioned above, the hypermethylation of *COSMC* resulted in the loss of function, and MUC2 expression was downregulated in both human CRC tissues and the CRC cell line LS174T [48]. In the study of pancreatic cancer cases, Prakash et al. found that hypermethylation of *COSMC* was the main cause of truncated Tn and STn O-glycan expression [54]. In addition, COSMC and C1GALT1 are also transcriptionally regulated by members of the specific protein/Krüppel-like transcription factor family. These genes can be seen as targets for the regulation of mucin-type O-glycan synthesis, but studies in this area are still relatively scarce [86].

## 6. Aberrant Mucin-Type O-Glycosylation in Gastric Cancer

Mucin-type O-glycans, mainly core 1 and core 3 derived O-glycans, constitute the main mucus barrier of the gastrointestinal tract, and their expression is usually found in non-neoplastic gastric tissues [87]. Normal gastric mucosa specifically express MUC1, MUC5AC and MUC6, but not MUC2 [88]. The expression levels of each gastric mucin may exhibit modest differences between different regions of the stomach [89]. For example, MUC1 and MUC5Ac are highly expressed in the superficial minor concave epithelium (MUC5Ac is also highly expressed in the mucus neck cells of the gastric sinus), whereas MUC6 is expressed in the gastric sinus as well as in the deeper glands [88].

Alterations in mucins are molecular markers of gastric mucosal malignancy. One of the major molecular events occurring during gastric carcinogenesis is the absence of mucin expression observed in a normal epithelium [90]. In intestinal-type gastric cancer, cancer cells exhibit abnormal mucin expression patterns characterized by the downregulation of MUC5Ac and MUC6, and the upregulation of MUC3, MUC4, and MUC5B expression [91]. Javanbakht et al. found that the mRNA levels of *MUC5Ac* were significantly lower in gastric cancer tissues by IHC and qRT-PCR compared with nongastric cancer tissues and that the low expression of MUC5Ac may be associated with the progression and poor prognosis of gastric cancer [92]. Numerous findings suggest that the reduced amount of the MUC5Ac protein in gastric cancer tissues is closely associated with *Helicobacter pylori* (*Hp*) infection and that MUC5Ac has a protective role in maintaining gastric sinus homeostasis, and inhibiting *Hp* infection colonization and associated inflammation [93]. However, chronic gastritis associated with *Hp* infection may evolve into intestinal metaplasia (IM), which can be classified into two categories based on histomorphological features: complete type, which is characterized by significant loss of the expression of MUC1; MUC5Ac; and MUC6, and accompanied by abnormal expression of MUC2; incomplete type characterized by the simultaneous expression of MUC1, MUC5AC, and MUC6, and accompanied by abnormal expression of MUC2; IM will further deteriorate and develop into gastric adenocarcinoma (Figure 3) [94,95].

In addition, the changes in glycosyltransferases are also important factors in the occurrence and development of gastric cancer. Previously, it was shown that GALNT6 and GALNT15 expression were significantly upregulated in gastric cancer tissues and the high expression of GALNT6 was significantly correlated with a low expression of E-cadherin and β-catenin, as well as high expression of MMP9 [96,97]. Moreover, GALNT10 could promote the proliferation and migratory ability of gastric cancer cells by enhancing the expression of HOXD13 and decreasing the sensitivity to 5-Fu [98]. On the other hand, the role of C1GALT1 in gastric cancer cannot be ignored. Dong et al. found that C1GALT1 promoted gastric cancer cell proliferation, migration, and invasion by regulating the O-glycosylation of integrin α5, and thus activated the PI3K/AKT pathway, identified integrin α5 as a novel downstream target of C1GALT1 by lectin pull-down assay and mass spectrometry analysis in gastric cancer [99]. High expression of C1GALT1 in gastric adenocarcinoma is associated with poor prognosis and poor survival. Both in vivo and in vitro experiments showed that C1GALT1 promotes EPHA2 phosphorylation and Ephrin A1-mediated cell migration by regulating the O-glycosylation of EPHA2, suggesting that C1GALT1 may be a potential therapeutic target for gastric cancer [100]. Surprisingly, human COSMC and C1GALT1 share a 26% amino acid sequence homology, yet the role of COSMC in the development of gastric carcinogenesis has not been reported so far [27].

## 7. Tumor Therapy Based on Abnormal Mucin-Type O-Glycosylation

In 1985, Springer reported the first mouse monoclonal antibody against the Tn antigen, and then Hakomori with his colleagues developed the first mouse monoclonal antibody against the STn antigen [101,102]. In the following years, many monoclonal antibodies against Tn and STn have been produced successively, such as CA3638, Tn antigen-specific chimeric monoclonal antibody (Chi-Tn), and B72.3 against STn antigen [103,104,105]. But the biggest problem with these antibodies is their lack of specificity. For example, many monoclonal antibodies against Tn antigen lack specificity. In addition to recognizing Tn antigen, they can also cross react with other GalNAc-containing glycans, such as type A blood [106].

Antibodies against Tn antigens tend to cross react with human IgA1 glycoforms. For this problem, Cummings and his team developed a set of new recombinant anti-Tn monoclonal antibodies, Remab6 and ReBaGs6, in 2020. Remab6 is a chimeric human IgG1 antibody and ReBaGs6 is a mouse IgM antibody. Both Remab6 and ReBaGs6 can recognize clustered Tn structures, but the most important thing is that they do not recognize IgA1. In flow cytometry and immunofluorescence analysis, Remab6 can recognize human cancer cell lines expressing Tn antigen, but not Tn negative control cells. In immunohistochemistry, Remab6 can stain cancer tissues, but rarely normal tissues. These data indicate that ReBaGs6 and Remab6 are promising tools for the detection, diagnosis, and treatment of human cancer [107].

Different antibodies, vaccines, and targeted drugs can be developed for different cancers. Posey, Jr., et al., developed and characterized a new CAR based on the background of the Tn-MUC1-specific monoclonal antibody 5E5 that inhibits tumor growth in mouse models of leukemia and pancreatic cancer. This suggests that epitopes based on aberrant glycosylation formation may serve as targets for CAR-T therapy [108,109]. In addition, mucin-related tumor markers have been widely used in clinical practice, for example, MUC16 (CA125) has been widely used as a marker for ovarian cancer; SLe^a^ (also called CA19-9) is a marker for pancreatic cancer; MUC1 (CA153) is a biomarker for breast cancer [110,111,112]. Considering that aberrant O-glycosylation plays an important regulatory role in tumors, TAG-72 was originally isolated from a xenograft of the human colon cancer cell line LS-174T by monoclonal antibodies binding of B72.3. The monoclonal antibody TAG-72 was found to recognize and bind antigenic epitopes on mucins, thereby inhibiting tumor-associated immune responses, and complete surgical excision of TAG-72-positive tissue reversed tumor immune escape and leading to improved overall survival in patients with primary or recurrent CRC [113]. MacLean et al., developed the vaccine against STn, which is composed of the STn antigen structure and was initially used to treat metastatic breast cancer. In clinical trials, patients who received the vaccine tolerated it well, had a significant improvement in survival of 12.1 months, and developed a humoral immune response against the STn antigen [114]. Vaccination is a promising therapeutic approach if the right antigenic marker is found to not only enhance the antitumor immune response but also to generate a lasting immune memory. The use of anticancer vaccines targeting mucin-type O-glycosylation is promising, however, it may induce immune intolerance. Therefore, people have used carbohydrate antigens covalently coupled to certain carrier molecules to form clustered or multiepitope-conjugated vaccines or to induce antibody-dependent cellular cytotoxicity (ADCC), using antibodies against specific targets expressed in tumor cells [115]. Lakshminarayanan et al. designed a glycosylated vaccine for a mouse breast cancer model consisting of a TLR1/2 ligand (Pam3CysSK4), a T-helper cell (Th) epitope, and an abnormal MUC1 peptide that produces cytotoxic T cells (CTL) that recognize both glycosylated and nonglycosylated peptides, whereas a similar nonglycosylated vaccine-generated CTL can only recognize nonglycosylated peptides. The antibodies elicited by the glycosylated tripartite vaccine are more catalytic than those of the nonglycosylated control, making immunization with the glycosylated tripartite vaccine more advantageous in preventing tumors [116]. Not coincidentally, Chang et al. in 2022 developed an STn-based adjuvant vaccine, which is a conjugate vaccine composed of clustered STn antigen (triSTn), TLR1/2 ligand (Pam3CSK4), and T-helper cell (Th) epitopes, which they found to be effective in producing anti-triSTn IgG antibodies. In addition, they used flow cytometry for the first time to analyze immune cells (T cells, B cells, dendritic cells, and other monocytes) in the spleens of mice injected with the vaccine to assess the effect of the vaccine and found that the vaccine was effective in activating immune cells and inducing specific immune responses leading to efficient antibody production. Notably, the vaccine did not cause excessive inflammatory responses in the body [117]. Guided nanomedicines are emerging therapies for oncology and have shown great potential to improve disease outcomes in preclinical and clinical drug development stages, Chen et al. conjugated gold nanoparticles (GNPs) with anti-MUC7 antibody as a probe to target urothelial cancer cells. They exposed the conjugated GNPs to a green light laser to kill transitional cell carcinomas (TCCs) and preserve normal cells, and found that nanoparticles conjugated with the MUC7 antibody could kill cancer cells at lower power. However, more animal studies are needed to confirm these findings [118]. 

## 8. Summary

Mucin-type O-glycosylation is a common posttranslational modification process of proteins, whose normal biosynthesis is inextricably linked to the maintenance of organismal health, and which is involved in various biological processes. In contrast, aberrant O-glycosylation is frequently accompanied by several diseases such as tumors, including the production of tumor-associated antigens. Tn, STn, and T antigen, and alterations in the activity or localization of some glycosyltransferases, are closely associated with tumor development; it has been well established that these alterations can modulate a range of malignant phenotypes of tumor cells and lead to reduced patient survival by inhibiting or activating signaling pathways. In the field of precision oncology, the belief that aberrant mucin-type O-glycosylation is a key factor in the complex molecular structural alterations of tumors highlights the potential of using mucin-type O-glycans as tumor-specific therapeutic targets and biomarkers. Altered glycosylation was discovered more than 60 years ago in malignant transformation and was considered to be one of the hallmarks of tumor pathogenesis. Since then, tumor therapies targeting them have been continuously explored, and a variety of monoclonal antibodies and vaccines have been generated against the abnormal mucin-type O-glycans. Although many problems still accompany them, they have shown potential in the diagnostic and therapeutic development of a wide range of cancers. With the advent of novel antibody engineering technologies, the focus of research has shifted to generating genetically engineered antibody fragments with higher binding affinity and tumor localization capabilities. We believe that the recent development of single-cell high-throughput sequencing, whole-exome sequencing, and nanoparticle technologies will provide a deeper understanding of the application of mucin-type O-glycans in tumor immunotherapy.

## Figures and Tables

**Figure 1 cells-11-03666-f001:**
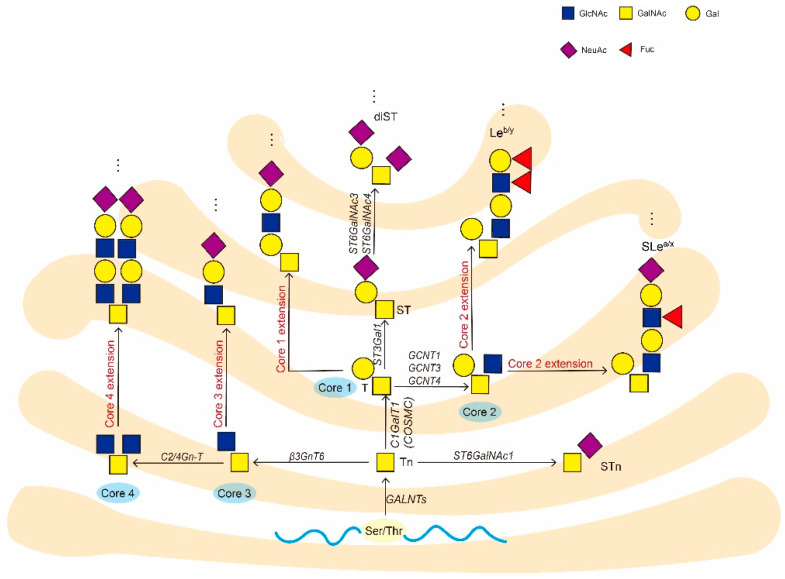
Biosynthesis of O-GalNAc glycans. The Ser/Thr-linked α-GalNAc O-glycan is ubiquitous in mammals and present on many different glycoproteins. The first step of O-GalNAc glycan synthesis is the linking of UDP-GalNAc to serine/threonine residues of proteins in the presence of GALNTs to form the initial structure GalNAcα1-O-Ser/Thr (Tn antigen). Next, the Tn antigen extends to synthesize either the core 1 structure via C1GALT1 or the core 3 structure via β3GnT6. Core 2 or core 4 structures can be further synthesized after core 1 or core 3 is formed. STn antigen is generated through ST6GalNAc-I by adding sialic acid on the α1-6 linkage of GalNAc residue of Tn. These core structures are usually further extended and terminated by sialic acid and fucose residues. Moreover, the most common modification of Tn antigen is the generation of T antigen in the presence of C1GALT1, which requires the collaboration of COSMC molecular chaperones for the exertion of C1GALT1 activity. Glycan structures are represented according to the nomenclature of glycan symbols (SNFG) format [28].

**Figure 2 cells-11-03666-f002:**
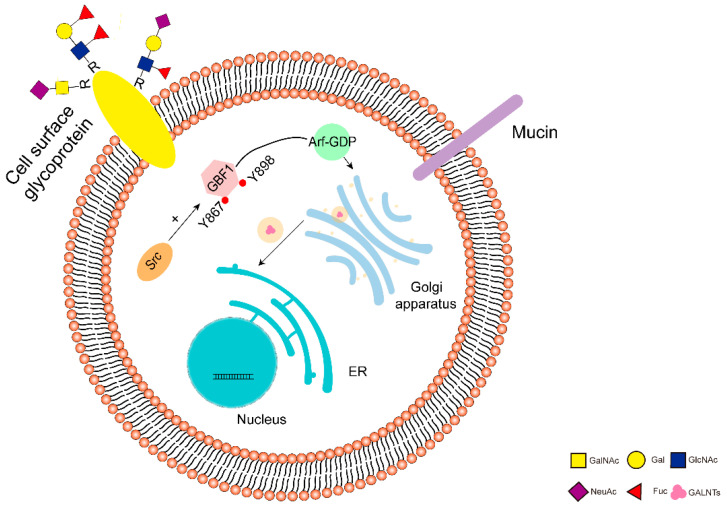
Src facilitates the relocalization of GALNTs from Golgi to ER by promoting the binding of GBF1 to Arf1. The transfer of GALNTs from Golgi to ER is called “GALA pathway”, which is essential for tumor growth. The transport of GALNTs from Golgi to ER is dependent on the GTP exchange factors GBF1 and Arf-GDP, Src induces the formation of tubular transport carriers containing GALNTs and also phosphorylates GBF1, where phosphorylation of two sites on GBF1, Y876, and Y898, and facilitates the binding of GBF1 to Arf-GDP. This led to the efflux of GALNTs from the Golgi membrane and their eventual relocalization to the ER.

**Figure 3 cells-11-03666-f003:**
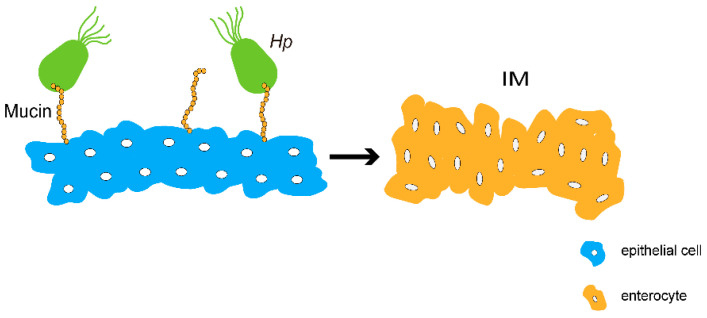
Abnormal mucin expression causes *Hp* to colonize the gastric epithelium, resulting in Intestinal epithelial metaplasia (IM) of the gastric mucosa. Mucins are highly glycosylated glycoproteins. Some mucins, such as MUC5Ac, are an important component of the gastrointestinal mucosal defense system, which protects the gastric surface from chemical, enzymatic, mechanical, and microbial damage. It is important that mucins carry glycan epitopes that provide binding sites for microorganisms, including the gastric oncogenic bacterium *Hp*.

**Table 1 cells-11-03666-t001:** The roles of glycosyltransferases and relative enzymes in different diseases involved in the text.

Generated Structures	Glycosyltransferases or Relative Enzymes	Types of Diseases	Impact on Diseases	Ref.
Tn antigen	GALNT1	Renal fibrosis	Suppression	[32]
	GALNT8	Breast cancer	Suppression	[75]
	GALNT3	Familial tumoral calcinosis	Suppression	[34]
	GALNT12	Glioblastoma multiforme	Promoting	[76]
	GALNT2	Non-small cell lung cancer	Promoting	[77]
	GALNT8	Breast cancer	Suppression	[78]
	GALNT2	Gastric adenocarcinoma	Suppression	[79,80]
T antigen	COSMC	Colorectal cancer	Promoting	[46]
		Proliferating hemangiomas	Promoting	[47]
		Colorectal cancer	Suppression	[48]
		Pancreatic cancer	Suppression	[81]
	C1GALT1	Colorectal cancer	Suppression	[45]
		Laryngeal carcinoma	Promoting	[42]
		esophageal cancer	Promoting	[43]
		Head and neck cancer	Promoting	[44]
STn antigen	ST6GalNAc-I	Prostate cancer	Promoting	[61]
		Breast cancer	Promoting	[62]
		Bladder cancer	Promoting	[63]

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
