# Peer review of "A Sweet Warning: Mucin-Type O-Glycans in Cancer"

_cells, 2022, doi:10.3390/cells11223666_

Round 1

Reviewer 1 Report

This manuscript “A sweet warning: mucin-type O-glycans in cancer” is introducing the relationship between mucin-type glycans and diseases, especially cancer, from a view of biosynthesis based on a lot of literatures. This review is very comprehensive and represents a massive effort of authors. I believe this paper is acceptable, but it would be more beneficial to the readers if the authors would consider the following point.

Comment on Figure 1-3:

I am afraid that the figure legends or the main text contain very little explanation for all figures, so that the reader cannot decipher the contents of the figures. The figure should be understandable just by looking at it and the captions (legends), and also everything shown in the figure should be properly explained.

Reviewer 2 Report

The review article is interesting and well written. My main criticism concern the lack of information about N-glycosylation of mucins and the occurence of shared carbohydrate antigens between N- and O-linked glycans in several glycoproteins including mucins. The authors should distinguish and make clear when mucin-type O-glycans are found on mucins and when on other glycoproteins.

Moreover, the wide list of roles of O-glycans and related glycosyltransferases and other enzymes should be organized and presented in a table where structures, enzymes, roles, and references are made easily recognizable by the readers.

Mispelling

page 9 line 376: ... Galbeta1-33[Fuc ... : change to 1-3[Fuc ..
